Maize plant height automatic reading of measurement scale based on improved YOLOv5 lightweight model

Li Jiachao 1 2 3
Zhou Ya’nan 1 2
Zhang He 1 2 4
Pan Dayu 1 2
Gu Ying 1 2
http://orcid.org/0000-0001-7007-2489 Luo Bin 1 2 3 luob@nercita.org.cn
1 Intelligent Equipment Research Center, Beijing Academy of Agriculture and Forestry Sciences , Beijing , China
2 National Engineering Research Center for Information Technology in Agriculture , Beijing , China
3 College of Mechanical and Electrical Engineering, Xinjiang Agricultural University , Urumqi, Xinjiang , China
4 College of Agriculture, Northeast Agricultural University , Harbin, Heilongjiang , China
Alatas Bilal
Electronic publication date: 2024 Aug 5
Publication date: 2024
Volume: 10
Electronic Location ID: e2207
Received 2023 Nov 16; Accepted 2024 Jun 27
Copyright: © 2024 Li et al.
Copyright year: 2024
Copyright holder: Li et al.
License: This is an open access article distributed under the terms of the Creative Commons Attribution License, which permits unrestricted use, distribution, reproduction and adaptation in any medium and for any purpose provided that it is properly attributed. For attribution, the original author(s), title, publication source (PeerJ Computer Science) and either DOI or URL of the article must be cited.
License URL: https://creativecommons.org/licenses/by/4.0/

Keywords: Plant height measurement, Deep learning, Neural network, Attention mechanism

Funding: National Key Research and Development Program of China 2022YFD2002301 National Natural Science Foundation of China 62273125 This work was supported by the National Key Research and Development Program of China (No. 2022YFD2002301) and the National Natural Science Foundation of China (No. 62273125). The funders had no role in study design, data collection and analysis, decision to publish, or preparation of the manuscript. The funders had no role in study design, data collection and analysis, decision to publish, or preparation of the manuscript.

==============================
Background

Plant height is a significant indicator of maize phenotypic morphology, and is closely related to crop growth, biomass, and lodging resistance. Obtaining the maize plant height accurately is of great significance for cultivating high-yielding maize varieties. Traditional measurement methods are labor-intensive and not conducive to data recording and storage. Therefore, it is very essential to implement the automated reading of maize plant height from measurement scales using object detection algorithms.

Method

This study proposed a lightweight detection model based on the improved YOLOv5. The MobileNetv3 network replaced the YOLOv5 backbone network, and the Normalization-based Attention Module attention mechanism module was introduced into the neck network. The CioU loss function was replaced with the EioU loss function. Finally, a combined algorithm was used to achieve the automatic reading of maize plant height from measurement scales.

Results

The improved model achieved an average precision of 98.6%, a computational complexity of 1.2 GFLOPs, and occupied 1.8 MB of memory. The detection frame rate on the computer was 54.1 fps. Through comparisons with models such as YOLOv5s, YOLOv7 and YOLOv8s, it was evident that the comprehensive performance of the improved model in this study was superior. Finally, a comparison between the algorithm’s 160 plant height data obtained from the test set and manual readings demonstrated that the relative error between the algorithm’s results and manual readings was within 0.2 cm, meeting the requirements of automatic reading of maize height measuring scale.

Introduction

A crop phenotype refers to the physical, physiological, and biochemical characteristics that reflect the cellular, tissue, organ, plant, and population structures and functions of plants. It represents the time sequence and three-dimensional expression of plant gene maps, as well as their geographical variations and generational evolutionary routines (Jiang & Li, 2020). Plant height is one of the most crucial phenotypical traits, significantly affecting crop growth, photosynthesis, lodging resistance, and mechanical harvesting, which play a key role in determining crop yield (Wang & Su, 2022). The plant height of maize is an essential agronomic trait related to the stem lodging resistance. It also serves as a vital indicator of maize phenotypic parameters. The phenotypic measurements of plant height can provide the important references for the quantitative analysis of genotype-environment interactions and contribute to maize breeding. Traditional methods for measuring plant height often involve using rulers, which are simple, portable, and cost-effective. However, this method may result in human-induced reading errors and require labor-intensive data recording and organization, and low detection efficiency.

Improvements in computer vision technology have introduced new methods for measuring plant height. The pixel information can be extracted from reference objects based on shape, color, or texture for crop height measurement. This information is then converted into height information using other reference objects or distance data, resulting in an accurate measure of plant height. For instance, Sritarapipat, Rakwatin & Kasetkasem (2014) placed a ruler of known height in a rice field and used image processing techniques such as filtering and threshold segmentation to capture the unobstructed part of the ruler, thus indirectly measuring the height of rice plants. Mano (2017) used white metal wire mesh as a reference to calculate rice plant height in the field. Li et al. (2022) used article as a reference and employed the skeleton extraction and shortest path computation methods to calculate the height of rice seedlings. However, the accuracy of measuring plant height using image processing methods depends on the acquired pixel information. When measuring taller crops from a longer distance, significant errors may arise. Moreover, these methods exhibit poor robustness in complex natural environments.

With the development of artificial intelligence and computer hardware, deep learning algorithms have become an important research topic and have been widely applied across various domains. Researchers have integrated deep learning methods into the study of crop phenotyping. For example, Zou et al. (2020) used the networks like R-CNN, YOLOv3, and RetinaNet to achieve maize ear detection and counting. Deng et al. (2021) integrated the feature pyramid network (FPN) into fast region-based convolutional neural network (fast R-CNN) to identify and count grains on single-branch rice ears. Dong et al. (2022) proposed an improved wheat ear detection method based on AsymmNet. Appe, Arulselvi & Balaji (2023) added the convolutional block attention module (CBAM) into YOLOv5 to detect tomato fruits. Fan et al. (2023) proposed a spatio-temporal convolutional neural network model that leveraged the shifted window transformer fusion region convolutional neural network model for the purpose of detecting pineapple fruits. However, existing methods (Bronson et al., 2021; Kim et al., 2021; Luo et al., 2021; Wang et al., 2023) have mostly relied on depth cameras, ultrasonic sensors, and similar devices for measuring crop height. These methods have significant measurement errors and come with high equipment costs, thus making them unsuitable for crop height measurements in field breeding experiments.

To address this issue, a new approach was proposed based on the traditional maize plant height measurement. In this article, an improved version of YOLOv5 combined with text recognition was introduced to achieve an automatic reading of the maize plant height measurement scale. This study conducted lightweight improvements on the original YOLOv5 to facilitate the use of the model on mobile devices. Additionally, the model’s detection performance was enhanced by incorporating attention mechanisms and other methods. The specific objectives of this study are as follows: (1) The lightweight improvement was performed by replacing YOLOv5’s backbone with MobileNetv3. (2) The model’s detection accuracy was enhanced by using the NAM attention mechanism module. (3) The scale readings for maize plant height were calculated by modifying the detect.py file combined with the Easyocr package for recognizing scale digits using the scale and fixed sleeve detected by the improved model. (4) The performance of the improved model in detecting scale readings under various lighting conditions was evaluated.

Materials and Methods

Image data collection

The experimental setup for this study consisted of a maize plant height measurement scale and a smartphone. The image collection took place at the experimental field within the Beijing Academy of Agriculture and Forestry Sciences (39°56′ N, 116°16′ E, altitude 60 m).

The images were captured on July 20, 2023 (sunny day) and July 24, 2023 (partly cloudy). Under different lighting conditions, including normal, backlit, and front-lit scenarios, images of the maize plant with the measurement scale were taken using a Huawei P50 smartphone. The measuring range of the ruler was 1.2–3 m, and the index value was 1 cm. The scale was positioned beside the maize plant under measurement, and the phone was fixed at a distance of approximately 10 cm from the scale. A total of 436 images were captured with a resolution of 2,736 × 3,648 pixels. After removing blurred images caused by focusing issues, 400 high-quality original images were selected, as illustrated in Fig. 1.

Figure 1 Experimental setup and acquisition of images.

Image data preprocessing

To address the challenges posed by limited sample size and the effects of different shooting environments on model recognition, the image augmentation techniques including brightness enhancement, contrast enhancement, and color enhancement were applied to the original images to expand the dataset. In order to distinguish them from the original images, the brightness, contrast, and color of the images were randomly enhanced from 1.1 to 1.5 times of the original values. The augmented images are shown in Fig. 2. Finally, the 1,600 images with various backgrounds were obtained as a dataset and saved in .jpg format. A manual labeling method was employed and the scale and fixed sleeve of the ruler was marked using the LabelImg annotation tool for gaining the accurate position information of scales and sleeves. In order to reduce the errors in the final scale reading, the minimum bounding box of the scale line should be marked when marking the scale. Consequently, the .xml files containing the names and positions of scale and fixed sleeve were obtained. These files were subsequently converted into .txt files conforming to the training requirements of YOLOv5. The above annotated dataset was randomly divided into training set, validation set, and test set in the ratio of 8:1:1.

Figure 2 Data enhanced image.

(A) Raw image. (B) Brightness enhancement. (C) Contrast enhancement. (D) Color enhancement.

Overall structure of YOLOv5

Object detection algorithms are broadly categorized into two types: single-stage detection algorithms and two-stage detection algorithms (Jiao et al., 2019). Single-stage detection algorithms include the YOLO series (Redmon et al., 2016), SSD (Liu et al., 2016), and others, while two-stage detection algorithms included Faster-RCNN (Ren et al., 2015), Mask-RCNN (He et al., 2017), and others. In comparison to two-stage and SSD algorithms, the YOLOv5 model can maintain the detection accuracy of existing models and significantly improve the detection accuracy, thus making it better for achieving real-time object detection. The YOLOv5 network model primarily consists of the input end, backbone network, neck network, and detection network. Utilizing CutMix data augmentation as a baseline, four images were randomly scaled, cropped, and arranged into a new photo by mosaic data augmentation at the input end (Yun et al., 2019). Mosaic data augmentation may enrich the dataset, improve network robustness, reduce required graphics processing unit (GPU) performance, and ultimately lower costs. The backbone network was responsible for scale and fixed sleeve feature extraction, which consisted of the Focus module, C3 module, which was an improvement of the BottleneckCSP module, the CBS module, and the spatial pyramid pooling (SPP) module. The Focus module sliced the input scale image and expanded channel dimensions to extract target features of scale and fixed sleeves. To address the issue of repeated gradient information during network optimization causing excessive computation, the batch normalization layer after residual output and activation function layers were removed to decrease computational complexity. The CBS module included 2D convolution, batch normalization, and the SiLU activation function. The SPP module extracted ruler features through different kernel sizes and overlaid them for feature fusion, to significantly enhance feature extraction efficiency. The neck network primarily accomplished the fusion of scale and fixed sleeve features and was composed of FPN and the path aggregation network (PAN). FPN performed data fusion through up-sampling from top to bottom to boost object detection performance. PAN also extracted feature information from top to bottom to obtain more accurate location information. Ultimately, the extracted features were fused, which enabled the neck network to have both stronger semantic features and more precise positioning capabilities. The YOLOv5 loss function consisted of three components: bounding box loss, class loss, and confidence loss.

Improved YOLOv5 with lightweight modification

Mobilenetv3 primarily comprised Bneck structures (Howard et al., 2019), as shown in Fig. 3. It utilized depthwise separable convolution, inverted residual structures with linear bottleneck, and SE attention mechanism modules for better model compactness and high recognition accuracy, making it more suitable for use with a mobile terminal. In this study, the backbone network of YOLOv5 was replaced with the Mobilenetv3 lightweight model, and the NAM attention mechanism module was introduced into the neck network. The modified network structure is shown in Fig. 4.

Figure 3 Mobilenetv3-bneck structure drawing.

Figure 4 Improved network structure diagram.

Compared to traditional convolution, depthwise separable convolution employed the techniques of depthwise convolution and pointwise convolution (Chollet, 2017), which reduced parameter computation of the MobileNet series, as depicted in Fig. 5.

Figure 5 Depth separation convolution structure diagram.

(A) Deep convolution. (B) Point-by-point convolution.

Depthwise convolution, also known as channel-wise convolution, assigned each convolutional kernel to a specific channel; after performing depthwise convolution, the number of generated feature maps remained the same as the number of input layer channels. However, this method did not effectively utilize the feature information from different channels at the same spatial position and pointwise convolution was needed to merge the feature maps, creating new ones. The pointwise convolution operation was like that of traditional convolution, and its convolutional kernel size was 1 × 1 × M, where M represented the number of channels in the previous layer. Pointwise convolution would combine the input feature maps from the previous step with weight in the depth direction, thus generating new feature maps. Here, DF represented the dimensions (length and width) of the feature map, DK denoted the size of the convolutional kernel, M represented the number of input feature map channels, and N represented the number of convolutional kernels.

The computational of depthwise separation convolution was as follows:

(1) D F2⋅M⋅D K2+D F2⋅M⋅N

The computational of traditional convolution was:

(2) D F2⋅M⋅N⋅D K2

The inverted residual network architecture was constructed by using 1 × 1 convolutions for dimensionality increase, followed by 3 × 3 depthwise convolutions for feature extraction, and 1 × 1 convolutions for dimensionality reduction, which was opposite to the traditional residual network structure and better utilized the convolutional kernel feature extraction function. The ReLu6 activation function caused feature information loss during high-dimensional to low-dimensional mapping, therefore, the hardswish was replaced with the ReLu activation function, which used a linear activation bottleneck structure at the end layer.

(3) Hardswish(x)=x⋅ReLu(x+3)6

(4) f(x)=max(0,x)

The squeeze-and-excitation (SE) attention mechanism module consisted of two main steps: squeeze and excitation (Hu, Shen & Sun, 2018). In the squeeze step, the input feature map underwent global pooling, resulting in a multidimensional vector that encapsulated global information. This vector’s dimensions matched the number of channels in the input feature map. The excitation step involved two consecutive fully connected operations on the compressed result. The first operation utilized the ReLU activation function, followed by the second operation using the Hardswish activation function, which could obtain a weight vector with the same dimension as the input feature map. Ultimately, the feature map was weighted channel by channel through multiplication to the previous feature.

Improvement of channel attention mechanism

The importance of each channel was assumed to be the same after convolution in YOLOv5. However, in practical scenarios, the importance among different modules often varied. Replacing the YOLOv5 backbone with the lightweight Mobilenetv3 backbone could lead to a decrease in the model’s detection accuracy. However, introducing attention mechanism modules into the neck network might enhance its accuracy. The NAM was a lightweight efficient attention module (Liu et al., 2021) designed to reduce the weight of less salient features. The NAM module applied a penalty based on weight sparsity, enabling it to maintain similar performance while achieving higher computational efficiency. The NAM module primarily consisted of two sub-modules: channel attention and spatial attention, as illustrated in Fig. 6. The feature map first underwent channel attention module, and its output was then used as the input for the spatial attention module, ultimately resulting in an enhanced feature map. The channel attention sub-module leveraged scaling factors from batch normalization, as described in Eq. (5):

(5) Bout=BN(Bin)=γBin−μBσB2+ε+β

where μB and σB were the mean and standard deviation of the mini-batch B, and γ and β were trainable affine transformation parameters. A larger scaling factor indicated greater variation in the channel, implying richer information content and higher channel importance. The channel attention sub-module was shown in Fig. 6A, where MC represented the output feature, γ was the scaling factor for each channel, and Wγ represented the associated weight. By applying the scaling factors of BN to the spatial dimension, the importance of pixels could be evaluated. The spatial attention sub-module was depicted in Fig. 6B, with MS denoting the output feature, λ representing the scaling factor for each channel, and Wλ being the corresponding weight. To suppress less important features, a regularization term was introduced in the loss function, as shown in Eq. (8).

(6) MC=sigmoid(Wγ(BN(F1)))

(7) MS=sigmoid(Wλ(BNS(F1)))

(8) Loss=∑(x,y)⁡l(f(x,W),y)+p∑g(γ)+p∑g(λ)

Figure 6 NAM attention mechanism module.

(A) Channel attention submodule (B) Spatial attention submodule.

YOLOv5 employed the complete intersection over union (CIoU) as the bounding box loss, although CIOU Loss considered the overlap area, center point distance, and aspect ratio of bounding box regression. However, the CIoU Loss reflected the difference in aspect ratio, rather than considering the individual differences between width and height with their corresponding confidences. As a result, this hindered the model’s effective optimization of similarity. To address this issue, some scholars have broken down the aspect ratio based on CIOU and proposed the extended intersection over union (EIOU) Loss. The penalty term of EIoU has separated the impact factor of aspect ratio and calculated the length and width of the target box and anchor box separately based on the penalty term of CIoU. This loss function comprised three parts: overlap loss, center distance loss, and width-height loss. The first two parts continued the methods from CIoU, while the width-height loss directly minimized the differences in width and height between target and anchor boxes, leading to faster convergence. The EIoU loss function was represented by Eq. (9). In this study, the EIoU would be used in place of CIoU to further enhance the accuracy of the model.

(9) LEIoU=1−EIoU=1−IoU+ρ2(b,bgt)c2+ρ2(ω,ωgt)Cω2+ρ2(h,hgt)Ch2

Recognition and calculation of scale readings

In this study, the EasyOCR library was employed for recognizing the graduations on the ruler. EasyOCR was an OCR library written in Python, designed to recognize digits, characters, and text within images, providing output in text format. EasyOCR first employed the CRAFT algorithm to identify text regions, followed by the utilization of convolutional recurrent neural networks (CRNN) for text recognition. The CRNN model comprised a feature extraction layer Resnet, a sequential modeling layer LSTM, and a decoding layer CTC.

Using the improved lightweight YOLOv5 model, pixel position information for the scale and fixed sleeve could be obtained. The vertical coordinate of the fixed sleeve reading position was recorded as y1, and the pixel distance between the unit scale vertical coordinate was recorded as y2. Through the EasyOCR library, it could be obtained that the number closest to the fixed ruler was d1, and the vertical coordinate of the number closest to the fixed sleeve was y3.

According to Eq. (10), the actual distance between the number closest to the fixed sleeve and the fixed sleeve was d2. Finally, the ultimate reading of the ruler, num, could be obtained using Eq. (11).

(10) d2=(y1−y3)/y2

(11) num=d1+d2

Model training and evaluation index

The model training platform used in this study was configured with an Intel (R) Core (TM) i5-12400 F 2.50 GHz CPU, 16 GB of RAM, a GeForce GTX 1660 SUPER GPU with 6 GB of VRAM. The operating environment was Windows 10 system, Python version of 3.9, PyTorch version of 1.12.1, and CUDA version of 11.3.

This study primarily employed precision (P), recall (R), and mean average precision (mAP) to reflect the training accuracy of the model. Parameters, computational complexity, and model weight size were used to indicate the model’s complexity. Frame per second (FPS) was utilized to demonstrate the real-time detection performance of the model. The P represented the proportion of correctly predicted samples among all samples, as shown in Eq. (12):

(12) P=TPTP+FP×100%

The R represented the proportion of correctly predicted samples among all positive samples, as shown in Eq. (13):

(13) R=TPTP+FN×100%

The mAP was the mean of the average precision (AP), where AP was the area under the precision-recall (P-R) curve, as shown in Eq. (14):

(14) mAP=1N∑i=1N⁡∫01⁡P(R)dR×100%

In the equation, TP represented the number of true positive predictions, FP represented the number of false positive predictions, FN represented the number of false negative predictions, and N represented the number of categories. In this study, we only discussed one category, the spore of rice blast, so N = 1. mAP@0.5:0.95 was used as the evaluation metric for the model. The mAP@0.5:0.95 referred to the average precision within the intersection over union (IoU) threshold range of 0.5 to 0.95. Specifically, it was the average value of AP calculated at IoU thresholds ranging from 0.5 to 0.95 with a step size of 0.05.

Results

Model training results

During training, the Adam optimization algorithm was utilized with image dimensions adjusted to 640 pixels × 640 pixels. The initial momentum, learning rate, initial weight decay coefficient, batch size, and training epochs were set to 0.937, 0.001, 0.1, 8, and 150, respectively. The model file with the highest accuracy and the last time after training were saved.

Training with the improved network architecture led to increasing precision, recall, and mean average precision as the number of training epochs increased. The final accuracy, recall rate, mAP50, and mAP@0.5:0.95 were 98.8%, 99.2%, 98.6%, and 69.2%, respectively. The memory footprint of the improved model was 1.8 MB, with a computational complexity of 1.2 GFLOPs. The real-time detection frame rate on the computer was 54.1 fps. These results indicated that the model not only maintained detection accuracy but also significantly reduced model size and computational complexity, which could meet the ultimate usage needs on the mobile end.

Ablation experiment of the improved network

Replacing the original backbone network with the lightweight Mobilenetv3 backbone led to a significant reduction in model size in the YOLOv5 network. However, both precision and mAP@0.5:0.95 showed a decline. The cause of this issue lied in the reduced number of model parameters after changing the backbone network, which hampered effective image feature extraction, subsequently diminishing the network’s feature extraction capability.

To enhance the model’s detection accuracy, attention mechanism modules were integrated into the network, and the CioU loss function was replaced with the EioU loss function. As shown in Table 1, substituting the YOLOv5 backbone with the Mobilenetv3 lightweight model backbone led to a model size of only 12.9% of the original network’s size. Adding the NAM attention mechanism modules and replacing the CioU loss function with the EioU loss function, improved the accuracy and mAP@0.5:0.95 by 0.4% and 1.2%, respectively, compared to the lightweight network. This was because the NAM (channel-wise) attention mechanism could enhance or suppress different channels based on the importance of each feature channel, thereby improving model performance. The EIOU loss function addressed the issue in the CIOU loss function where the penalty term was always zero when the aspect ratios of the predicted box and the ground truth box were the same. This modification contributed to the enhancement of model performance. In comparison to the original network, the improved network exhibited a substantial reduction in model size with a slight decrease in accuracy and mAP@0.5:0.95. The loss curves of the improved model on the training and validation sets were illustrated in Fig. 7. The bounding box regression loss curve, target confidence loss curve, and class classification loss curve of the improved model converged quickly and eventually stabilized (Fig. 7). This indicated that the improved model could accurately identify targets of different categories and precisely localize them, meeting the detection requirements.

Table 1 Ablation test result.

Basic model	Mobilenetv3	Attention module	EIoU	Precision/%	Recall/%	mAP/%	mAP@0.5:0.95/%	Model size/MB	
YOLOv5s	—	—	—	99.1	99.2	98.9	71.3	13.9	
✓	—	—	98.4	99.0	98.5	68.0	1.8	
✓	✓	—	98.6	99.1	98.6	68.8	1.8	
	✓	✓	✓	98.8	99.2	98.6	69.2	1.8	

Figure 7 Loss function curve.

Discussion

Comprehensive comparison experiment of different detection models

The results of the improved lightweight model with YOLOv5s, YOLOv7 and YOLOv8s models are shown in Table 2.

Table 2 A comparative test of different attention mechanisms.

Model	Precison/%	Recall/%	mAP50/%	Calculated amount/GFLOPs	Detection speed/fps	Model size/MB	
YOLOv5s	99.1	99.2	98.9	15.8	32.8	13.9	
YOLOv7	99.1	99.3	99.1	105.1	8.6	71.3	
YOLOv8s	98.8	98.9	98.9	28.4	50.5	21.4	
Ours	98.8	99.2	98.6	1.2	54.1	1.8	

Based on the experimental results in Table 2, it could be observed that various network models exhibited high precision, recall, and average precision values, with little variation. This indicated that the maize plant height measurement ruler dataset created in this study had good quality. The YOLOv7 detection model achieved the highest detection accuracy, but it had the largest model size and computation complexity and the slowest detection speed, making it unsuitable for deployment on mobile devices. YOLOv8 was the latest model in the YOLO series. In comparison to the original YOLOv5s model, the YOLOv8s model showed an increase in computation complexity and model size, a decrease in detection accuracy, and an improvement in detection speed. There were fewer network layers in the YOLOv8s model compared to the YOLOv5s model, resulting in faster feature extraction and fusion, but reduced accuracy. Using the detection models from Table 2 to analyze the same test set, a randomly selected image under side light, back light, and front light conditions was presented in Fig. 8.

Figure 8 Comparison of different model detection.

The results indicated that under different lighting conditions, the improved lightweight model, like other detection models, could accurately identify the positions of ruler scales and sleeves without any instances of missed detection. The YOLOv8s network model exhibited poorer confidence scores, and coupled with its relatively high detection accuracy, and this might be attributed to overfitting issues during the training process, resulting in suboptimal performance on the training set.

According to the comparative experiments and model recognition performance, it was evident that the proposed lightweight model in this study had the smallest computational complexity and model size, the fastest detection speed, and a higher detection accuracy. It could accurately detect ruler scales and sleeves without instances of missed detection. In summary, the proposed lightweight model exhibited the best overall performance and met detection requirements with high speed, making it suitable for deployment on mobile devices.

Plant height measurement experiment

The text and position of the numbers on the ruler were obtained using Python’s EasyOCR library using the proposed model to obtain the scale of the ruler and the position information of the fixed sleeve. The scale readings could be calculated by modifying the YOLOv5’s detect.py file, ultimately getting the plant height information of maize. The improved model accurately identified scale readings under different lighting conditions, as depicted in Fig. 9.

Figure 9 Test results of RBS-YOLO model in different environments.

A comparison was made between the plant height data obtained for 160 samples through the algorithm from the test set and compared with the manual reading to verify the accuracy of the plant height measurements. The fitted outcome of the predicted and measured values through linear regression for the plant height measured by scale is shown in Fig. 10.

Figure 10 Correlation between manual and algorithmic readings.

It is evident from Fig. 10 that there was a significant linear relationship between the predicted values obtained by algorithm and measured values of the plant height measured by scale. The correlation and determination coefficient between the predicted and measured values were 0.9994 and 0.9996, respectively. This indicated a close relationship between the true and measured values, demonstrating that the algorithm proposed in this study met the requirements for plant height measurement.

Conclusions

This study established an image dataset for the plant height measurement scale and proposed a lightweight model based on YOLOv5 to automate the reading of height rulers for maize. The original backbone network was replaced with the MobileNetv3 network. The NAM attention mechanism module was introduced into the neck network, and the EioU loss function replaced the CioU loss function. The effectiveness of the improved model was verified through ablation experiments. The average accuracy of the improved network model was 98.6%, the mAP@0.5:0.95 was 69.2%. The computational complexity was 1.2 GFLOPs, the model occupied 1.8 MB of memory, and the computer detection frame rate was 54.1 fps, which had good analysis accuracy and efficiency.

The scale and fixed sleeves on the ruler were detected with this improved model. Python’s EasyOCR library was used to recognize the numbers on the ruler, and the rewritten detect.py was used to calculate the ruler’s reading to obtain the plant height of maize.

The results indicated that the proposed maize plant height measurement model could meet the real-time and high-precision requirements for field applications when the algorithm-generated data were compared with the manually collected data. Consequently, this algorithm may be more convenient for breeding experts conducting field experiments. In the future, we will collect images of different types of measuring scales, expand the data set, improve the model, select different measuring scales according to different crops, and extend the proposed algorithm to the measurement of plant height of more types of crops.

Supplemental Information

Supplemental Information 1 Data profile.

Supplemental Information 2 Model configuration file.

Supplemental Information 3 Detection code.

Additional Information and Declarations

Competing Interests

Author Contributions

Data Availability

The authors declare that they have no competing interests.

Jiachao Li conceived and designed the experiments, performed the experiments, analyzed the data, performed the computation work, prepared figures and/or tables, authored or reviewed drafts of the article, and approved the final draft.

Ya’nan Zhou conceived and designed the experiments, performed the experiments, prepared figures and/or tables, authored or reviewed drafts of the article, and approved the final draft.

He Zhang performed the experiments, prepared figures and/or tables, authored or reviewed drafts of the article, and approved the final draft.

Dayu Pan analyzed the data, prepared figures and/or tables, authored or reviewed drafts of the article, and approved the final draft.

Ying Gu performed the computation work, prepared figures and/or tables, authored or reviewed drafts of the article, and approved the final draft.

Bin Luo conceived and designed the experiments, prepared figures and/or tables, authored or reviewed drafts of the article, and approved the final draft.

The following information was supplied regarding data availability:

The data is available at figshare: Li, Joish (2023). Maize plant height measuring scale data set. figshare. Figure. https://doi.org/10.6084/m9.figshare.24547165.v1.

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
