# Peer review of "Maize plant height automatic reading of measurement scale based on improved YOLOv5 lightweight model"

_PeerJ Computer Science, doi:10.7717/peerj-cs.2207_

## Round 0.1 · original submission · Major Revisions

The paper needs to be revised to strengthen its writing structure and clarify its interesting topic for the reader. The contribution needs to be more clarified. The results and comparison with existing techniques require more strengthening and thorough discussion. All the reviewers' comments are highly important to be considered and fixed.

Reviewer 1 has suggested that you cite specific references. You are welcome to add it/them if you believe they are relevant. However, you are not required to include these citations, and if you do not include them, this will not influence my decision

**Language Note:** The review process has identified that the English language must be improved. PeerJ can provide language editing services - please contact us at [email protected] for pricing (be sure to provide your manuscript number and title). Alternatively, you should make your own arrangements to improve the language quality and provide details in your response letter. – PeerJ Staff

Reviewer 1 ·

Basic reporting

This paper proposes a lightweight detection model based on improved YOLOv5s to achieve automatic reading of the maize plant height measurement scale. It utilizes a combination of algorithms to achieve this goal, and the test results demonstrate that the algorithmic results and the relative error of the manual readings are within 0.2 cm. However, the current document has several weaknesses that must be addressed in order to obtain a documentary result that is equal to the value of the publication.

Experimental design

(1)In the abstract section, it is recommended to include a comparison of the improved performance of the model in the results, highlighting the superiority of the improved model.
(2)The paper covers several evaluation indicators. It would have been appropriate for the authors to provide a brief explanation of them.
(3)It seems that the author may have forgotten to include the serial numbers of the formulas after the revision. Formulas 6 and 11 mentioned in the text were not found.
(4)It is suggested to conduct testing experiments for both the ablation experiments in this paper and the various model comparison experiments, and then present the results of the inspection and testing, which I believe will be more convincing.

Validity of the findings

(5)The paper should mention the scope for further research as well as the implications/application of the study.
(6)The authors may add more state-of-art application articles for the integrity of the manuscript (A Performance Analysis of a Litchi Picking Robot System for Actively Removing Obstructions, Using an Artificial Intelligence Algorithm; Agronomy. Transforming unmanned pineapple picking with spatio-temporal convolutional neural networks. Computers and Electronics in Agriculture. Path planning for mobile robots in unstructured orchard environments: An improved kinematically constrained bi-directional RRT approach; Computers and Electronics in Agriculture.).

Reviewer 2 ·

Basic reporting

1. The manuscript needs to be revised from some writing issues.
2. What is the purpose of measuring plant height except that machinery tools.
3. It is recommended for the separation of the introduction section for the discussion of related work, with distinct sections allocated for the introduction and related work.
4. Figure 4 two times mentioned, in lines 140 and 144. The second time should be a different figure, according to the text.
5. In section Recognition and Calculation of Scale Readings:
o Equation 10 has mentioned again in line 232 , does the same equation work for loss function and calculation of scale reading at the same time.
o Equation 11 is mentioned in line 234 but is missing .
6. In section Model training and evaluation index:
o There is a lack of mathematical background for evaluation metrics .
7. In section Ablation Experiment of the Improved Network:
o The text “ As shown in Table 1, substituting the YOLOv5 backbone with the Mobilenetv3 lightweight model backbone led to a model size of only 12.9% of the original networks size.” In lines 270-272 is not clear based on table 1 values .

Experimental design

1. In Introduction section. I suggest that you improve the description at lines 78-80 to provide more explanation for your study (specifically, you should explain upon the knowledge of the dataset being used). Does it mean the dataset created through testing the model?

2. In section, Materials AND Methods:
a) The dataset employed in this manuscript has been found to lack an optimal explanatory capacity based on the available information. Kindly provide a detailed elaboration on the missing points, including the following aspects:
o Specify the duration or creation time of the dataset.
o The dataset size in terms of the number of records (rows).
o Details about the image collection process, including factors like distance and environmental conditions.
o The process of expanding the dataset through data augmentation lacks clarity, particularly regarding the sample numbers. It is recommended to show the sample numbers before and after augmentation using a Table.
b) It is recommended to add an original YOLOv5 network architecture in section Overall Structure of YOLOv5
c) In the section "Improved YOLOv5 with Lightweight Modification," details about the improved model, such as the kernel number and kernel size of each layer, were not provided.

3. In section Improvement of Channel Attention Mechanism
o In a normalization-based attention module, channel attention and spatial attention mechanisms were used separately or merged together on your proposed model, as shown in figure 6(a),6(b).

4. Based on the values showed in Table 1, the improved model only reduced model weigh size whereas the accuracy result decreased.

5. In section Comprehensive Comparison Experiment of Different Detection Models:
o The clarity of the statement in lines 282-283, " The ShuffleNet-v5 model replaced the backbone network of YOLOv5s with the ShuffleNet network " is ambiguous. It remains unclear whether the ShuffleNet-v5 model serves as a comparative model, similar to SSD, or if it is used as a backbone for YOLOv5.
o Convergence the loss functions for both training and validation sets of the improved YOLOv5 should be presented.

Validity of the findings

1. Does it make sense if we are going to use NAM more than one block in the neck network section. It is used in the last layer of the proposed model as shown in figure 4.
2. In section Ablation Experiment of the Improved Network:
o Two other metrics such as recall and mAP@50) were missed in Table 1, whereas they were mentioned in section Model training and evaluation index at line 240-242.
3. It is recommended to use the original YOLOv5s backbone and replace other sections such attention module and loss function. For enhanced accuracy results coupled with a reduction in weight size, as illustrated in Table 1.

4. Row Data
o I thank you for providing the raw data, however your supplemental files need more descriptive metadata such as model yaml and common file.

Additional comments

This article has an interesting research topic but needs more improvement in the structure. A revision of this article is needed to improve its writing structure and make it more effective in conveying its an interesting idea to the reader.

Annotated reviews are not available for download in order to protect the identity of reviewers who chose to remain anonymous.

Reviewer 3 ·

Basic reporting

This article aims to improve the performance of the yolov5 network with the help of several improvements.
we are glad to review this paper. However, we have some questions and suggestion for the authors to improve their manuscript.

1. The writing in the manuscript needs improvement as there are numerous mistakes that diminish its readability. The authors are advised to carefully review the manuscript and rectify all errors.

2. The article lacks a discussion on interpretability. Despite achieving high accuracy, the proposed method may not be easily understood by readers who seek insight into the factors influencing the classification decision. Sufficient discussion on enhancing the interpretability of the proposed method is needed.

3. The presented individual improvements in this article do not represent a novelty with regard to yolo network. Rather, it is an approach to modelling the network with different setups that include already existing impovements. Unfortunately, this objective was not adequately explained in the article.

4. The main contributions should be summarized.

5. In the introduction, why do you select the Yolov5 as the main primary network to improve?

6. In conclusion, more discussions should be made regarding the limitation and future dimensions of the proposed study.

Experimental design

1. In terms of experimental design, although the authors have accomplished surpassing the existing methods in several evaluation metrics, there are still many problems with the experimental design of this article.

2. What is the data setup for this experiment (train, val, test?)

3. There are no visual comparisons of the data set.

4. Have you compared the models on other public datasets?

5. For K-means++, CA, SE, and CBAM, these modules are used frequently in today's yolov5 improved models, so why were not they chosen? What is the main difference?

6. The authors should have emploed cross-valiation for the evaluation method.

7. Have you compared with other new modules proposed in last 1-2 years? not the yolov5 series.

8. Are images taken in the morning, midday and afternoon with constantly changing shooting angle and distance. The captured images include complex and changeable conditions such as maize plant occlusion, maize plant overlap occlusion, dense target maize plant , branch occlusion, back light, front light, side light and other maize plant scenes?

Validity of the findings

no comment

---

## Round 0.2 · accepted · Accept

Dear authors,

Thank you for the revision and for clearly addressing all the reviewers' comments. In the light of this revision, it appears that your paper has been improved and is now acceptable for publication.

Best wishes,

Reviewer 1 ·

Basic reporting

accept

Experimental design

accept

Validity of the findings

accept

Additional comments

accept

Reviewer 2 ·

Basic reporting

The authors have revised the manuscript and addressed all comments. However, they still need to ensure accurate adherence to the PeerJ template, especially regarding text and figure names, as mentioned.

Experimental design

No cooment

Validity of the findings

No comment

Additional comments

Thank you for providing the raw data. However, the supplemental files require the common.py file.